# Transcriptional Regulatory Network of Plant Cadmium Stress Response

**DOI:** 10.3390/ijms24054378

**Published:** 2023-02-22

**Authors:** Yakun Li, Lihong Ding, Mei Zhou, Zhixiang Chen, Yanfei Ding, Cheng Zhu

**Affiliations:** 1Key Laboratory of Specialty Agri-Product Quality and Hazard Controlling Technology of Zhejiang Province, College of Life Sciences, China Jiliang University, Hangzhou 310018, China; 2Department of Botany and Plant Pathology, Purdue University, West Lafayette, IN 47907, USA

**Keywords:** cadmium stress, transporter, transcription factors, regulatory network, plants

## Abstract

Cadmium (Cd) is a non-essential heavy metal with high toxicity to plants. Plants have acquired specialized mechanisms to sense, transport, and detoxify Cd. Recent studies have identified many transporters involved in Cd uptake, transport, and detoxification. However, the complex transcriptional regulatory networks involved in Cd response remain to be elucidated. Here, we provide an overview of current knowledge regarding transcriptional regulatory networks and post-translational regulation of the transcription factors involved in Cd response. An increasing number of reports indicate that epigenetic regulation and long non-coding and small RNAs are important in Cd-induced transcriptional responses. Several kinases play important roles in Cd signaling that activate transcriptional cascades. We also discuss the perspectives to reduce grain Cd content and improve crop tolerance to Cd stress, which provides a theoretical reference for food safety and the future research of plant varieties with low Cd accumulation.

## 1. Introduction

Cadmium (Cd) is one of the naturally occurring heavy metals, which is extremely toxic to plants and humans [1]. In recent years, the increase in Cd content in soils has caused serious and widespread pollution to farmland. The accumulation of Cd in plants has toxic effects on the normal growth of plants. For example, Cd affects enzyme activity and the absorption and consumption of essential elements, generates reactive oxygen species (ROS), and impairs photosynthesis, respiration, and membrane systems. All these effects ultimately result in plant tissue necrosis, chlorosis, and eventual death [2,3]. Cd is also a threat for human health. The bone itai-itai disease in Japan in the 1950s was caused by long-term consumption of rice (*Oryza sativa* L.) produced in Cd-contaminated soils [4]. Cd enters the human body through the food chain and mainly accumulates in the kidneys, causing a series of diseases, such as anaemia, cancer, heart failure, steoporosis, emphysema, and renal function diseases [5,6,7,8]. Therefore, it is necessary to limit Cd in the food chain from soils to reduce health risks to humans.

Recent studies have made important progress in elucidating the physiological and molecular mechanisms of Cd transport and tolerance in plants. According to the relation between the metal content in the soil and metal in the plants, plants are divided into three groups: excluder, indicator, and hyperaccumulator plants [9]. So far, many transporters related to Cd uptake, transport, sequestration, and detoxification in plants have been identified [10,11,12,13] (Table 1, Figure 1). Metal transporters and ROS-scavenging enzymes are major functional proteins that are induced by Cd stress. Heavy metal accumulation and tolerance in plants are associated with a highly complex regulatory network system involving a large number of genes. Recent studies in rice, *Arabidopsis thaliana*, and other plants have revealed multi-layered transcriptional networks comprising many transcriptional factors (TFs), long non-coding RNAs (lncRNAs), and microRNAs (miRNAs) in responses to Cd stress [14,15,16] (Figure 2). An increasing number of reports indicate that epigenetic regulation, such as DNA methylation, is important in Cd-induced transcriptional responses. Many kinases play important roles in Cd signaling that activate transcriptional cascades [17,18]. In this review, we focus on recent findings regarding the transcriptional network and post-translational regulation of TFs that control the expression levels of metal-responsive genes. The review on the regulatory mechanisms of Cd uptake, transport and accumulation in plants is of great significance for reducing Cd content in food crops to ensure food safety.

## 2. Cd Transport and Accumulation in Plants

Cd transport and accumulation in plants have been most extensively investigated in rice and involve four steps: (1) uptake by roots, (2) xylem-loading-mediated translocation to shoots, (3) redistribution through stems and nodes, and (4) further translocation to grains through the phloem [4]. As shown in Table 1, many metal transporters related to Cd uptake, transport, and detoxification have been cloned in plants, including iron (Fe)-regulated transporter1 (*OsIRT1*) [20,54], *OsIRT2* [54,55], natural resistance-associated macrophage protein 1 (*OsNRAMP1*) [33], *AtNRAMP1*, *AtNRAMP3*, *AtNRAMP4* [28,29,30,31], zinc (Zn)-/iron-regulated transporter-like protein 1 (*OsZIP1*) [21,22], *OsZIP3* [21,23], Cd accumulation in leaf 1 (*CAL1*) [51], *OsNRAMP5* [35,56], *HvNRAMP5* [32], cation/calcium (Ca) exchanger (*OsCCX2*) [38], heavy metal ATPase 2 (*OsHMA2*) [42,57], *OsHMA3* [43,58], low-affinity cation transporter 1 (*OsLCT1*) [46], and oligopeptide transporter 3 (*OPT3*) [53]; excluders–ATP-binding cassette, subfamily C/G (*OsABCG36*) [49], pleiotropic drug resistance 8 (*AtPDR8*) [50], and plant cadmium resistance protein 2 (*SaPCR2*) [52]. The discovery of these genes provides an important theoretical and practical basis for molecular breeding of crops with low Cd accumulation.

### 2.1. Cd Entry into the Roots

#### 2.1.1. Cd Absorption by Transporters

At present, there are no transporters in plant roots that specifically absorb Cd. Cd can be absorbed mainly through synergistic action by other essential mineral elements, such as Zn, Fe, and manganese (Mn) ions. Several metal transporters, like OsIRT1, OsNRAMP1, and OsNRAMP5 have been reported to be responsible for Cd entry into rice roots [33,54]. OsIRT1 and OsIRT2 are located on the plasma membrane. After 10 days of 100 μM CdSO_4_ treatment, the expression of *OsIRT1* and *OsIRT2* was highly increased in rice roots [59]. OsIRT1 and OsIRT2 both display Cd, Fe, and Zn influx activities in yeast, and overexpression of *OsIRT1* increases these metals in different plant tissues [20]. These results indicate that both OsIRT1 and OsIRT2 play important roles in rice by uptaking Cd from soil to roots. OsCd1, a major facilitator superfamily (MFS) protein, has been demonstrated to be involved in Cd uptake in root cells. OsCd1 resides in the plasma membrane of roots and contributes to Cd accumulation in rice grains [48]. In addition, OsNRAMP5 is located in the plasma membrane and is mainly responsible for the transport of Cd, Fe, and Mn in the rice root system. The *osnramp5* knockout mutants significantly reduced Cd concentration in roots and buds and increased Cd tolerance [11,60,61]. OsNRAMP1 is highly homologous to OsNRAMP5 and is also involved in Cd uptake and transport by root cells. Knockout of *OsNRAMP1* resulted in a significant decrease in the uptake of Cd and Mn by rice roots [33]. These results have important implications for the application of *OsNRAMP1* and *OsNRAMP5* mutations in mitigating Cd toxicity and reducing the risk of Cd contamination in rice grains. Under 10 μM CdCl_2_ stress for three days, compared with the yeast transformed with an empty vector, the growth of yeast expressing *AtNRAMP1*, *AtNRAMP3*, and *AtNRAMP4* was seriously impaired. In the meantime, after 3 μM CdCl_2_ treatment for 24 h, these three kinds of yeasts contained more Cd in yeast cells than the yeast transformed with empty vector [29]. Under 2 μM CdSO_4_ stress for 14 days, the growth of the *nramp1 Arabidopsis* mutant roots was little affected compared to the wild type (WT) [28]. Under both 1 and 10 μM CdCl_2_ stress for 10 days, the growth of *AtNRAMP3* overexpression in *Arabidopsis* roots was significantly reduced compared to WT [29]. Under 500 nM CdCl_2_ treatment for 14 days, the *AtNRAMP4* overexpression in *Arabidopsis* roots accumulated more Cd than WT [31]. These results indicate that these genes are related to Cd transport in *Arabidopsis* roots. HvNRAMP5, located in the plasma membrane, is also a major transporter for the uptake of Cd and Mn in barley [32].

#### 2.1.2. Cd Efflux by Transporters

OsZIP1 functioned as a metal-detoxified transporter through preventing excess Cd and Zn accumulation in rice [22]. OsZIP1 is located at the endoplasmic reticulum and plasma membrane [62]. *OsZIP1* overexpression in rice grew better under 5 μM CdCl_2_ stress for six days, but accumulated less Cd in plants. By contrast, the *oszip1* mutants and RNA interference (RNAi) lines accumulated more Cd in roots and displayed Cd-hypersensitive phenotypes. Tian et al. (2019) [23] found that both roots and shoots of *OsZIP3*-overexpressed transgenic rice plants were longer than those of WT plants under 10 μM CdSO_4_ for seven days. *OsZIP3* overexpression also reduced the Cd content in the roots and shoots. In addition, OsABCG36 localized at the plasma membrane was also involved in Cd efflux in rice roots. Knockout of *OsABCG36* increased Cd accumulation in root cell sap and enhanced Cd sensitivity [49]. SaPCR2 is localized at the plasma membrane and plays an important role in Cd detoxification. Under 15 and 30 μM CdCl_2_ stress for seven days, the Cd content in the roots of *SaPCR2*-overexpressed transgenic *Arabidopsis* plants were decreased compared to WT. Under 10 μM CdCl_2_ stress for seven days, the Cd content in the roots of *SaPCR2*-overexpressed transgenic *Sedum alfredii* plants were also decreased compared to WT plants. That means *SaPCR2* provided a route for Cd efflux in both *Arabidopsis* and non-hyperaccumulating ecotype (NHE) *S. alfredii* [52]. AtPDR8 is localized at the plasma membrane and was expressed in *Arabidopsis* roots and leaves. Kim et al. [50] found that under 5, 10, 20, and 30 μM CdCl_2_ for two to three weeks, *atpdr8* knockout plants and *atpdr8* RNAi plants were more sensitive to Cd than WT, while *AtPDR8*-overexpressed plants were resistant to Cd. That means AtPDR8 acts as an efflux pump of Cd^2+^ in plants.

### 2.2. Cd Transport to Shoots by Loading into the Xylem

Cd is transported to shoots by loading into the xylem vessel. Xylem-mediated root-to-shoot translocation is shown as a major determinant for shoot Cd accumulation in many plants including rice [63,64]. *CAL1* was a major quantitative trait locus (QTL) for Cd accumulation in rice leaves. CAL1 protein reduced Cd accumulation in rice leaves by specifically chelating Cd in the cytosol and promoting Cd secretion to extracellular spaces. *CAL1* also regulated Cd root-to-shoot translocation through the xylem, and *cal1* knockout mutants significantly reduced Cd concentration in rice leaves after 10 μM CdCl_2_ treatment for seven days [51]. *OsHMA2* is mainly expressed in rice roots and enriched in the vascular tissues, facilitating root-to-shoot Cd translocation. Knockout of *OsHMA2* significantly reduced Cd accumulation in shoots and grains [57]. The expression of *OsHMA2* was prominent in rice, which accumulated more Cd in its grains [65]. These results mean Cd can be transported from shoots to the xylem and, finally, to grains through OsHMA2. OsHMA3, a close homolog of OsHMA2, is a tonoplast-localized transporter for Cd in rice roots and is responsible for sequestering Cd in vacuoles [43]. Overexpression of *OsHMA3* significantly reduced Cd transport from roots to shoots and Cd content in grains (≥90%) [58,66]. Even in seriously Cd-contaminated soils, overexpression of *OsHMA3* alone produced rice grains with Cd concentration below the Chinese limit (Cd, 0.2 mg kg^−1^) [67], representing an ideal target for breeding low grain Cd rice. In *Arabidopsis*, the P1B-type ATPases, AtHMA2 and AtHMA4, both regulate root-to-shoot translocation of Cd and Zn and were mainly expressed in the vascular tissues of roots, stems, and leaves [39]. Overexpression of *AtHMA4* led to an increased tolerance to Zn, Cd, and Co and accumulated more metals in stems than WT [41]. Another P-type ATPase family member, AtHMA3, located at the vacuolar membrane, also participates in the vacuolar storage of Cd. Under 30 μM CdCl_2_ stress for 11 days, the roots and shoots of *AtHMA3*-overexpressed transgenic *Arabidopsis* plants accumulated more Cd than WT [40]. These results suggest that AtHMA3 plays a role in the detoxification of Cd through the vacuolar sequestration.

### 2.3. Cd Transport through the Phloem to Grains

Cd transported from the xylem to the shoots in rice is stored in nodes, transferred to the phloem, and then transported to rice grains through leaves, especially flag leaf phloem [48,68]. Phloem mediates nearly 100% of Cd deposition into grains in rice [69]. Cd can also be transferred to the grains through phloem in other plants, such as peanut (*Arachis hypogaea* L.), linseed (*Linum usitatissimum* L.), and potato (*Solanum tuberosum* L.) [70,71]. Cd mediated by phloem in *S. alfredii* participated in Cd remobilization from the older to younger leaves [72]. This reallocation could avoid excessive accumulation of Cd in leaves and stems. OsLCT1 is the first identified transporter for phloem Cd transport in plants [46]. *OsLCT1* is mainly expressed in leaf blades and nodes during the reproductive stage. The *Oslct1* knockdown mutant significantly reduced Cd content in rice grains as well as in phloem sap [46]. The expression of *OsLCT1* was significantly enhanced in rice, which over-accumulated Cd in grains, indicating possible translocation of Cd from shoots to grains [65]. These results suggest that *OsLCT1* in leaf blades functions in Cd remobilization by the phloem. In addition, OsCCX2, a putative Ca exchanger, is a node-expressed transporter involved in Cd accumulation in the grains of rice. *OsCCX2* is mainly expressed in the xylem region of vascular tissues at the nodes and plays a crucial role in mediating Cd translocation and distribution. Knockout of *OsCCX2* resulted in a significant decrease in Cd accumulation in rice grains when planted in 3.89 mg kg^−1^ Cd-contaminated paddy soils [38]. More recently, Gu et al. (2023) [69] identified a defensin-like gene, DEFENSIN 8 (*DEF8*), as the phloem Cd unloading transporter. *DEF8* is mainly expressed in rice grains. The *DEF8* mutant significantly decreased Cd accumulation in rice grains, offering an effective strategy to reduce the risk of Cd contamination without affecting important agronomic traits or the concentration of essential micronutrients. OPT3 is located at the plasma membrane and preferentially expressed in the *Arabidopsis* phloem. After 50 μM CdCl_2_ stress for two weeks, the *OPT3*-overexpressed transgenic *Arabidopsis* plants reduced the accumulation of Cd in grains and the *opt3* mutant *Arabidopsis* plants accumulated more Cd in grains and roots [53]. These results suggest that *OPT3* plays an important role in the transport of Cd from phloem to grains.

## 3. The Transcriptional Regulation of Cd Response by TFs

Recent studies have identified the complex transcriptional networks of plant Cd stress responses (Figure 2). TFs are major regulators of plant growth and development, as well as in abiotic and biotic stress responses. TFs belong to different families, such as WRKY, myeloblastosis protein (MYB), basic leucine zipper (bZIP), and heat shock transcription factor (HSF) [73,74,75]. They play important roles in signal transduction of Cd stress response by activating or repressing a series of genes involved in Cd uptake, transport, and tolerance in rice. The sensing of heavy metals by plants generates responses such as modulation of molecular and biochemical mechanisms of cells [58,76]. The ultimate plant Cd stress responses include altered synthesis of metal transporter proteins and metal binding proteins to counteract excessive metal stress in plants [74,77].

### 3.1. WRKY

The WRKY family is a unique plant TF family and plays an important regulatory role in plant development and response to various environmental stresses [75]. Under Cd stress, 35 WRKY genes were differentially expressed in rice, of which 25 were up-regulated and 10 were down-regulated. Under Cd treatment, the expression of *OsWRKY15* was induced in both leaves and roots, which may participate in Cd response via NO and ABA signaling pathways. The expression of *WRKY104* increased more than 90-fold after 24 h of Cd treatment [78]. Under Cd stress, *WRKY12* negatively regulated Cd tolerance via the glutathione (GSH)-dependent PC synthesis pathway in *Arabidopsis*. *WRKY12* directly targeted *GSH1* by binding to its promoter and indirectly inhibited the expression of other PC synthesis-related genes (*GSH1*, *GSH2*, *PCS1*, and *PCS2*), thereby negatively regulating Cd accumulation and tolerance in *Arabidopsis* [79]. The expression levels of *TaWRKY74* were significantly induced by Cd stress in wheat. *TaWRKY74* alleviated Cd toxicity in wheat by regulating the expression of Ascorbic Acid (ASA)-GSH synthesis genes [16]. In addition, Cd stress induced the expression of *WRKY13*. Overexpression of *WRKY13* decreased Cd accumulation and enhanced Cd tolerance, while the loss of function of *WRKY13* led to Cd accumulation and increased Cd sensitivity. WRKY13 can bind the promoter of the Cd extrusion pump gene *PDR8* and activate its expression to positively regulate Cd tolerance in *Arabidopsis* [80].

### 3.2. MYB

The MYB TF family is a large and functionally important class of proteins involved in the regulation of diverse biological processes. MYB proteins are divided into four classes according to the number and position of MYB repeats: 1R-MYB/MYB-related, R2R3-MYB, R1R2R3-MYB, and 4R-MYB [81]. *BnMYB2*, encoding a 1R-MYB protein from *Boehmeria nivea* (ramie), was significantly up-regulated in roots and leaves under Cd stress. The overexpression of *BnMYB2* in *Arabidopsis* resulted a significant increase in Cd tolerance and accumulation [82]. In addition, Tiwari et al. (2020) [83] identified another member of the rice 1R-MYB family involved in heavy metal tolerance. *OsMYB-R1*-overexpressed rice plants exhibited a higher auxin accumulation and a significant increase in lateral roots, which resulted in the increased tolerance under 150 μM and 300 μM Chromium (Cr) (VI) exposure for 21 days. RNA-seq analysis revealed over-representation of salicylic acid (SA)-regulated genes in *OsMYB-R1*-overexpressed rice plants [83]. These results imply that *OsMYB-R1* is part of a complex network of TFs controlling the cross-talk of auxin and SA signaling, which regulates heavy metal response.

The R2R3-*MYB* genes are more prevalent in plants and involved in regulating responses to environmental stresses [82,84]. Recent reports have established the role of *OsMYB45* in rice tolerance to Cd stress (Figure 3). The expression of *OsMYB45* was induced by Cd stress and highly expressed in the leaves, husks, stamens, pistils, and lateral roots of rice. Under 5 μM CdCl_2_ treatment for three days, the *Osmyb45* mutant was hypersensitive to Cd, which is associated with increased accumulation of hydrogen peroxide (H_2_O_2_) and reduced expression of antioxidative enzymes compared with WT. Catalase (CAT) is the main antioxidant enzyme and is encoded by three genes in the rice genome (*OsCATA*, *OsCATB*, and *OsCATC*). *OsCATA* and *OsCATC* expression was inhibited in *Osmyb45* mutations, which may be associated with Cd-sensitive phenotypes. The overexpression of *OsMYB45* in the mutant complemented the mutant phenotype [85]. In addition, another R2R3-type MYB member, MYB49a, was reported to be involved in the regulation of Cd accumulation in plants by physically interacting with the central ABA signaling molecule ABI5 [14]. *MYB49* was induced under Cd stress. Overexpression of *MYB49* in *Arabidopsis* significantly increased Cd accumulation, whereas *myb49* knockout plants reduced Cd accumulation. Further investigations revealed that MYB49 positively regulated the expression of basic helix-loop-helix (*bHLH*) TFs, *bHLH38* and *bHLH101*, by directly binding to their promoters and indirectly up-regulating expression of the *IRT1* transporter gene. MYB49 also binds to the promoter regions of the heavy metal-associated isoprenylated plant proteins, HIPP22 and HIPP44, leading to the activation of their expression and subsequent Cd uptake and accumulation [14].

### 3.3. bZIP

The bZIP family is one of the largest TF families in plants with important regulatory roles in various biological processes, including plant defense and responses to environmental challenges [86,87,88]. RNA-Seq results indicated that three differentially expressed genes encoding *bZIP6*, *bZIP19*, and *bZIP43* were involved in Cd stress in bentgrass [89]. After 400 μM CdCl_2_ treatment for four days in *Sedum plumbizincicola*, the expression levels of 32 *SpbZIP* genes changed and most of their expression levels peaked earlier in roots than in stems and leaves [88]. These results suggest that SpbZIP may play a major role in the initial response to Cd stress in the roots. In addition, TGACGTCA *cis*-element-binding protein (TGA) factors in *Arabidopsis* represent a subfamily of bZIP TFs. In *Arabidopsis*, TGA3 transcription was induced by Cd [90]. Compared with WT plants, the *tga3* mutant accumulated higher amounts of Cd in the roots and lower amounts in the shoots [91]. Fusco et al. (2005) [92] found that under Cd treatment, *BjCdR15*, acting as orthologue of TGA3 in *Arabidopsis*, regulated the expression of several metal transporters in *Brassica juncea*, such as *PDR8*, *HMA4*, and *NRAMP3*, thus mediating long-distance root-to-shoot transport of Cd. Overexpression of *BjCdR15* in *Arabidopsis* and *Nicotiana tabacum* (tobacco) enhanced Cd tolerance and accumulation in shoots [91]. These results indicate that bZIP TFs play crucial roles in the regulation of Cd accumulation, which provide useful candidates for potential biotechnological applications in the phytoextraction of Cd-contaminated soils.

### 3.4. HSF

The HSF family is an important member in plant stress response to several abiotic stresses by regulating the expression of stress-responsive genes, such as heat shock proteins (Hsps). In *Arabidopsis*, the Hsfs family is systematically divided into three classes of HsfA, B, and C [93]. In the plant response network, *HsfA1* specifically interacts with *HsfA2* to mediate the expression of genes encoding molecular chaperone HSPs such as HSP70 and HSP90 [94]. HSFs have been reported to play crucial roles in Cd tolerance in plants. *HsfA1a* conferred Cd tolerance in *Solanum lycopersicum* (tomato) by partially up-regulating *Hsps* expression [95] (Figure 4). After 100 μM CdCl_2_ stress for 15 days, *Hsfa1a*-silenced plants exhibited reduced melatonin levels, while *HsfA1a* overexpression stimulated melatonin accumulation and the expression of the melatonin biosynthetic gene caffeic acid O-methyltransferase 1 (*COMT1*). Exogenous melatonin promotes the modulation of GSH and PC biosynthesis which can detoxify Cd under Cd stress [96].

In *S. alfredii*, *SaHsfA4c* also played an important role in Cd tolerance. Compared with WT, the accumulation of ROS in *SaHsfA4c*-overexpressed *Arabidopsis* was reduced, and the expression of ROS-scavenging enzyme genes and *Hsps* was increased [97]. It has been found that the *TaHsfA4a* gene confers strong Cd tolerance in yeast and rice. *CUP1*, which encodes metallothioneins (MTs), contributes to the *TaHsfA4a*-induced Cd tolerance by acting as a downstream target of HsfA4a. OsHsfA4a is a rice homolog of TaHsfA4a. In rice plants expressing *TaHsfA4a*, Cd tolerance was enhanced, but in *oshsfa4a* knockdown rice plants, Cd tolerance was decreased. In addition, *TaHsfA4a* mediated Cd resistance in yeast by regulating MTs. The expression levels of *HsfA4a* and the *MT* gene were increased in rice roots under Cd stress. Therefore, HsfA4a in rice induced Cd tolerance by up-regulating *MT* gene expression in plants [98,99] (Figure 4).

### 3.5. Other TFs

The no apical meristem (NAM), *Arabidopsis* transcription activation factor (ATAF1/2), and cup-shaped cotyledon (CUC2) (NAC) family is a kind of pivotal TF in the response to various abiotic stresses [100]. They contain a conserved N-terminal DNA-binding NAC domain and a highly variable C-terminal domain. In *Aegilops markgrafii*, *AemNAC2* was found to be associated with reducing accumulation of Cd. Overexpression of *AemNAC2* could decrease accumulation of Cd in roots, shoots, and grains of transgenic wheat. In this type of transgenic wheat, *AemNAC2* suppressed the expression of *TaNRAMP5* and *TaHMA2* [101].

The ethylene responsive factor (ERF) family belongs to APETALA2/ethylene responsive factor (AP2/ERF) superfamily, which is one of the largest group of TFs involved in abiotic stress response in plants [102]. In *Glycyrrhiza uralensis*, overexpression of *lrERF061* led to maximum Cd uptake and enhanced antioxidant enzyme activities (SOD, CAT, and POD) under 10 mg L^−1^ Cd treatment [103]. This study contributes to the understanding of the role of *LrERF061* in Cd resistance and offers a useful way to increase the phytoextraction efficiency of Cd-polluted soils.

## 4. Regulation of Cd Response by DNA Methylation, Long RNAs, and Small RNAs

As discussed above, TFs are the core regulators of transcription under Cd stress. However, increasing evidence has revealed a complex regulatory system comprising not only TFs, but also DNA methylation, long RNAs, and small RNAs with crucial roles in Cd response (Figure 2).

### 4.1. DNA Methylation

Heavy metal stress has an effect on DNA structure, DNA stability, DNA methylation, and the regulation of gene expression. When these effects occur in plants, changes in DNA methylation can make plants adapt to heavy metal stress, especially to Cd stress [18,104]. DNA methylation can regulate gene expression and induce corresponding phenotypic changes without altering DNA sequence [105].

Cd treatment can lead to an increase in DNA methylation levels in rice, *Arabidopsis*, *Zostera marina*, and barley, which endows plants with higher tolerance to Cd [106,107,108,109]. Feng et al. (2016) [110] used high-throughput single-base-resolution bisulfite sequencing (BS-Seq) and RNA-Seq to analyze DNA methylation patterns in Cd-treated rice seedlings. A group of genes encoding metal transporters, Cd-detoxified proteins, and metal-related TFs were found to be differentially methylated, implying their roles in regulating rice tolerance to Cd stress. After 80 μM CdSO_4_ for four days, both *GSH2* and *GSHU35* upstream regions were hypermethylated. The sequence downstream of the coding region for iron-related transcription factor 2 (*OsIRO2*, a bHLH TF gene) was hypermethylated, while the coding region of metal transporter *OsZIP1* was hypomethylated. The expression level of *OsIRO2* was repressed, while *OsZIP1* was induced by Cd. These results suggest that DNA methylated modification was most likely involved in transcriptional regulation of metal transporter genes. Sun et al. (2022) [111] found that grafting significantly reduced the total sulfur and Cd accumulation in soybean, which was mediated by DNA methylation. The expression level of methyltransferase genes decreased, leading to the decreased expression of sulfur metabolism-related genes, especially S-adenosylmethionine (*SAM*). These results imply that DNA methylation was involved in a decrease in total sulfur and Cd content. In addition, Cd treatment can lead to a decrease in DNA methylation levels in *Trifolium repens* and *Cannabis sativa*, which reduces the tolerance of plants to Cd [112]. These results indicate that DNA methylation dynamics in response to Cd vary with species.

### 4.2. lncRNAs

LncRNAs are a class of non-protein coding RNAs with >200 nt, which act as ‘biological regulators’ to control transcriptional regulation and genome imprinting [106,113]. Many lncRNAs in plants were induced or inhibited by Cd stress, affecting plant morphology, physiology, and biochemistry, and thus producing response to stress. They were reported to play key roles in controlling the uptake of heavy metals by the plant system in order to minimize the uptake of heavy metals from soil to plants [15,114].

Chen et al. (2018) [115] used deep sequencing to study the differential expression of lncRNAs under Cd stress in rice. A total of 75 lncRNAs were down-regulated and 69 lncRNAs were up-regulated by Cd treatment. Analysis of the target gene related pathways revealed significant changes in genes associated with the cysteine (Cys) and methionine (Met) metabolic pathways, for example, Os03g0196600, which was involved in these pathways, was clearly up-regulated and might contribute to the production of Cys-rich peptides. XLOC_086307, the lncRNA targeted Os03g0196600 in cis, was also up-regulated significantly, which suggests that XLOC_086307 likely participated in Cd response in rice by regulating the Cys-rich peptide metabolism-related gene Os03g0196600. In addition, Feng et al. (2016) [106] identified 301 Cd-responsive lncRNAs in *Brassica napus* by RNA-seq analysis, of which 67 acted as competing endogenous target mimics (eTMs) for 36 Cd-responsive miRNAs. Four lncRNAs were identified to serve as precursors of miR824, miR167d, miR156d, and miR156e in response to Cd stress. Interestingly, TCONS_00035787 was shown to target miR167d in *B. napus*. The target gene of miR167d encodes a NRAMP1-type metal transporter, which plays an important role in Cd uptake in plants [106,116]. This is the first report of a lncRNA (TCONS_00035787)–miR167-*Nramp1* pathway in plants, indicating that lncRNAs can serve as new transcripts involved in the regulation of Cd uptake and accumulation in plants.

### 4.3. miRNAs

MiRNAs are a new class of small non-coding RNA molecules in plants, which negatively regulate specific target mRNAs at the post-transcriptional level. They are involved in plant growth and development, organ morphogenesis, and responses to heavy metal, drought, and chilling stress [117,118]. In our lab, Ding et al. (2011) [119] used miRNA microarray to analyze miRNA expression patterns in 60 μM Cd-treated and untreated rice seedlings. In addition to the up-regulation of miR528 under Cd stress, miR166, miR171, miR159, miR390, and miR192 were significantly inhibited [119,120]. Most of these miRNAs were reported to target TF genes, for example, miR166, miR171, and miR396 target homeodomain-leucine zipper TFs, scarecrow-like TFs, and growth regulating factor TFs, respectively. These results imply that miRNAs are key components of the transcriptional regulatory network of heavy metal stress responses in plants. The expression of miR166 was significantly repressed under 60 μM CdCl_2_ exposure in rice seedlings. Overexpression of miR166 reduced both Cd translocation from roots to shoots and Cd accumulation in the grains. In 35S: miR166 plants, the expression of *OsHMA2* decreased. Thus, the reduced Cd translocation in plants overexpressing miR166 may be at least partly attributable to the effect on *OsHMA2* expression [121]. In addition, miR390 was found to be significantly down-regulated under Cd stress. Overexpression of miR390 increased Cd accumulation and reduced tolerance to Cd toxicity in rice [122].

Meng et al. (2017) [123] found that miR167 could cleave *BnNRAMP1b* (one of the *NRAMP* genes), thus *BnNRAMP1b* was a target of miR167. Huang et al. (2010) [124] validated that miR395 targeted the sulfate assimilation related genes-sulfate transporter 2; 1 (*SULTR2*; *1*) and ATP sulfurylases (*APS*) by using 5′-RACE assay in *B. napus*. After 40 μM CdCl_2_ treatment for seven days, miR395-overexpressed *B*. *napus* plants exhibited high Cd accumulation and fewer toxicity symptoms in comparison to WT, due to increased synthesis of sulfur-containing compounds used for heavy metal chelation [125]. These results demonstrate the role of miR395 in the detoxification of Cd in *B. napus*. MiR398 targets two closely related cuprums (Cu)/Zn-SODs (CSDs), *CSD1* and *CSD2*, which promote defense against ROS accumulation in *Arabidopsis*. Transgenic *Arabidopsis* plants overexpressing a miR398-resistant form of *CSD2* accumulated more *CSD2* miRNA than plants overexpressing a regular *CSD2* and were consequently much more tolerant to heavy metals and other oxidative stresses [126]. Wang et al. (2022) [127] used high-throughput sequencing to analyze miRNA expression patterns in Cd-tolerant/sensitive barley. MiR156g was identified to be Cd-induced and target nucleobase-ascorbic acid transporters 2 (*HvNAT2*). *HvNAT2* was negatively regulated in the high-Cd-accumulating and Cd-tolerant genotype Zhenong8. Overexpression of *HvNAT2* enhanced ROS enzyme activities and GSH content, thus enhancing Cd tolerance in barley. These results indicate that metal-regulated miRNAs and their target genes are involved in the diverse processes of Cd response, including metal uptake and transport, sulfate allocation, metal chelation, and ROS detoxification.

## 5. How Plants Sense and Transduce Cd Signals to Transcriptional Regulators

How plants sense and transduce Cd signals to transcriptional regulators is one of the most important open questions. Recent studies revealed that heavy metal stress activates Ca^2+^ and ROS signaling that mediate signal transduction and enhance the expression of stress-responsive genes or TFs. ROS can also act downstream of the mitogen-activated protein kinase (MAPK) pathway [128]. MAPKs are among the most important and highly conserved signaling molecules that are activated by ROS production and induced upon metal stress. MAPK cascade consists of three tier components MAP kinase kinase (MAPKKKs/MEKKs), MAP kinase kinase kinase (MAPKKs/MEKs), and MAPKs/MPKs mediating phosphorylation reactions from the upstream receptor to the downstream target [129]. It has been shown that Cd stress activates different kinase enzymes belonging to the MAPK family. The phosphorylation cascade is therefore thought to be involved in Cd signaling to the nucleus. Research confirms that transcripts for OsMSRMK2 (OsMPK3 homolog), OsMSRMK3 (OsMPK7 homolog), OsBWMK1 (or OsMPK12), and OsWJUMK1 (OsMPK20-4 homolog) increased in response to Cd and Cu treatment in rice roots and leaves [130,131].

A connection between miRNA and MAPK signaling was deciphered by a study which showed regulation of miR398b/c by oxidative signal-inducible kinase 1 (*OXI1*) upon Cd and Cu treatment [132]. *OXI1* can enhance MAPK3 and MAPK6 activities based on the finding that knockout mutant plants for *OXI1* could not activate MAPK3 and MAPK6 under H_2_O_2_ treatment [133]. MEKK1 and ANP1 are both *Arabidopsis* MAPKKKs, which are regulated by H_2_O_2_ under Cd stress and can activate MAPK3 and MAPK6 through MKK4 or MKK5 [134]. Apart from this, several TFs, like bZIP-, MYB-, and myelocytomatosis (MYC)-related TFs, are known to act as downstream targets of MAPKs [135,136]. In addition, Opdenakker et al. (2012) [137] reported that downstream signal transduction targets of MAPK during Cd or Cu stress included WRKY22, WRKY25, and WRKY29. MAPK cascades regulate gene transcription by activating or inhibiting TFs such as WRKY and TGA (a subfamily of bZIP TFs), thus regulating a variety of cellular responses [138,139].

## 6. Future Perspective

Cd accumulation and exposure in crops poses a serious threat to organisms and human health. Breeding of new cultivars with low Cd levels is the most cost-effective and eco-friendly strategy to reduce the risk of Cd contamination in plants. To achieve this goal, we need a comprehensive understanding of not only the mechanisms but also the regulation of Cd uptake, translocation, sequestration, and other processes important for plant Cd stress responses. Over the past decades, different families of Cd transporters have been identified in plants, and their functional analysis through molecular and genetic approaches has provided critical insights into Cd uptake and translocation mechanisms (Figure 1). More recently, a large number of regulatory proteins including those involved in protein phosphorylation have been identified. Regulatory RNAs and DNA modifications have also been identified with roles in plant Cd accumulation and tolerance likely by affecting their expression, synthesis, activities, stability, and other properties (Figure 2). TFs are the core regulators of transcription under Cd stress. Several TFs in the transcriptional network and their functions during Cd stress have been analyzed. In addition, there is emerging evidence that epigenetic regulation through DNA methylation, lncRNAs, miRNAs, and kinases are involved in Cd-induced transcriptional responses. These signaling and responding mechanisms at transcriptional and post-transcriptional levels will facilitate our understanding of regulatory pathways and serve as a basis for developing efficient strategies to reduce Cd in plants.

Despite the important progress, our understanding of the signaling and complex transcriptional regulatory networks in Cd stress response remains to be very limited. First, it is unclear how plants sense Cd. Do plant cells sense Cd through specific recognition of Cd itself or through indirect recognition of certain Cd-associated molecules or induced effects? Given that all identified Cd transporters also transport other metal ions, it is possible that plants sense Cd simply as a heavy metal and there are overlapping mechanisms in signaling upon exposure to different types of heavy metals. Second, upon Cd stress perception, what are the earliest signaling events? Even though MAPK cascades are implicated in plant Cd signaling and responses, there are usually other regulatory proteins that act upstream of MAPK cascades. For example, in plant immune responses, plasma membrane-localized pattern-recognition receptors can recognize specific pathogen elicitors to trigger plant immune response through activation of the MAPK cascade. Given that Cd is transported into plant cells through plasma membrane-localized transporters, it is possible that the early signaling in Cd response starts at the plasma membrane as well and could directly involve Cd transporters through coordination with other proteins such as plasma membrane-localized receptor-like proteins. Third, even though a substantial number of TFs have been identified with a role in Cd accumulation and tolerance, many lack information about their regulation and action mechanisms. For example, it is unclear how some of the identified TFs are activated or induced in response to Cd exposure. For many TFs, this remains unclear regarding direct target genes under their regulation. More importantly, there is little knowledge about the cooperation and coordination among different TFs for the effective and tight control of the transcription programs of plant Cd responses. Fourth, more recent discoveries about the role of DNA methylation and regulatory RNAs in Cd responses will expand the complex transcriptional landscape of plant Cd stress responses. It will be critical to identify the target genes that are subjected to regulation by epigenetic mechanisms and regulatory RNAs and establish the processes and pathways by which these target genes influence plant Cd accumulation and responses. Finally, most of the research on plant Cd accumulation and responses has been carried out in rice and *Arabidopsis*. It is very likely that there are many unknown components and mechanisms that are present in different plants with important roles in plant Cd accumulation and tolerance. There are, for example, plants that hyperaccumulate Cd and can be highly valuable research materials for discovery of novel mechanisms by which plants accumulate, sequester, and detoxify high levels of Cd from heavily contaminated soils. In the hyperaccumulator *S. alfredii*, some genes related to Cd uptake and hyperaccumulation have been characterized, such as *SpHMA3*, *SaNramp6*, and *SaHsfA4c* [97,140,141]. Isolation of new genes including those TFs and interacting factors with regulatory roles in plant Cd accumulation and tolerance will help elucidate regulatory mechanisms in response to heavy metal stress. They can also be exploited as potential targets for genetic engineering through molecular breeding and clustered regularly inter-spaced short palindromic repeat (CRISPR)-Cas9 technology to reduce grain Cd accumulation and increase Cd tolerance in crop plants.

## Figures and Tables

**Figure 1 ijms-24-04378-f001:**
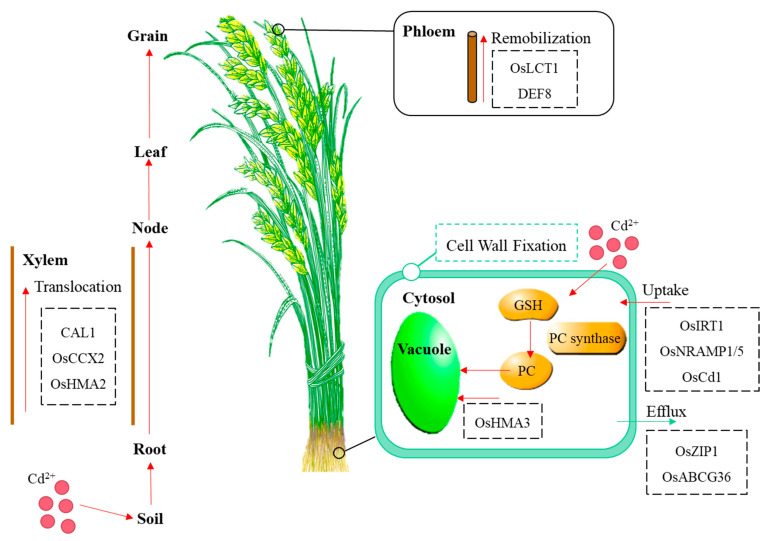
Transporters involved in the absorption and transport of Cd from soils to grains in rice. The pathways of Cd absorption by roots include Cd absorption and efflux by transporters, Cd fixation by the cell wall, and Cd chelation by vacuoles. Cd is transported to shoots by loading into the xylem. Then, Cd will be redistributed through stems and nodes and further translocated to grains through the phloem.

**Figure 2 ijms-24-04378-f002:**
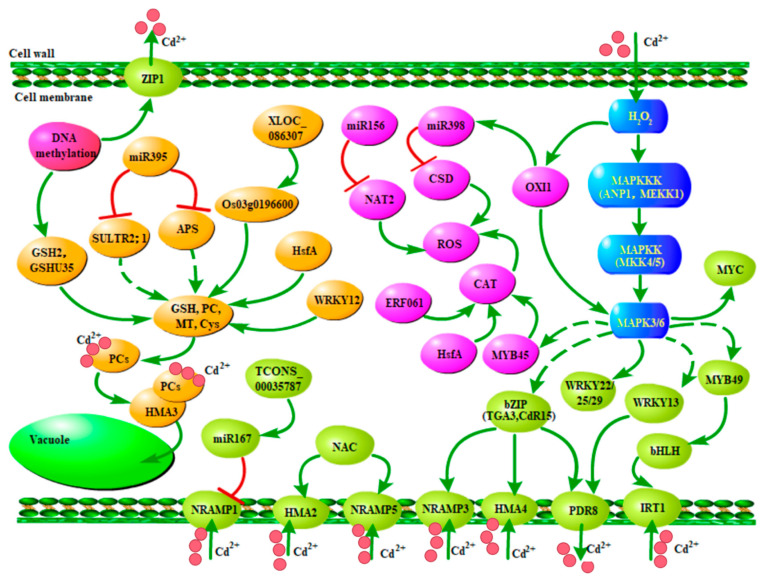
Overview of the transcriptional regulatory network in the response to Cd stress. As indicated in blue, H_2_O_2_ acts as a signaling molecule activating MAPK cascades. Key factors including TFs and metal transporters involved in Cd transport and efflux are indicated in green. Key factors involved in Cd chelation into vacuoles are indicated in orange. Key factors involved in ROS scavenge are indicated in purple. Arrows show simultaneous effects in the pathway, while the nail shapes represent repression. Dashed lines denote links to be confirmed.

**Figure 3 ijms-24-04378-f003:**
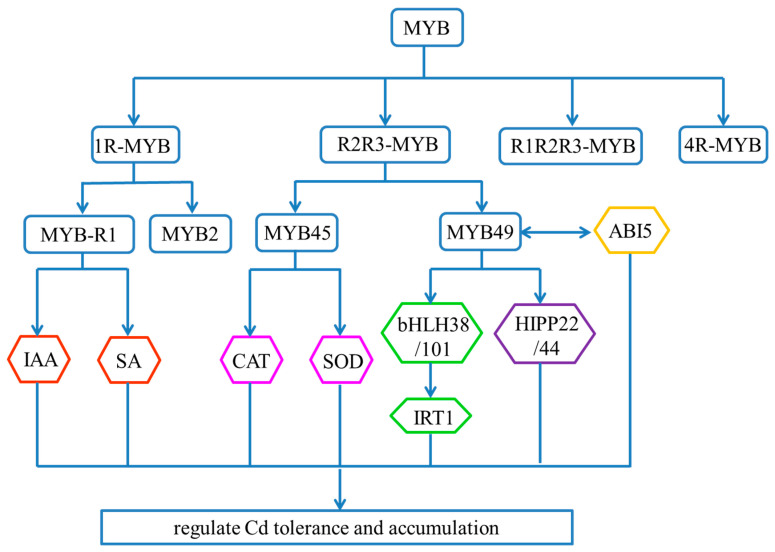
Transcriptional regulatory pathways of MYB TFs. MYB proteins are divided into four categories: 1R-MYB, R2R3-MYB, R1R2R3-MYB, and 4R-MYB. MYB2, as a 1R-type MYB, was up-regulated significantly under Cd stress. MYB-R1 is crucial for controlling the cross-talk of auxin and SA signaling and heavy metal response. The R2R3-MYB proteins are more prevalent in plants including MYB45 and MYB49. MYB45 regulates Cd tolerance and accumulation by producing CAT and SOD. MYB49 has three ways to regulate Cd tolerance and accumulation: (I) up-regulates the expression of metal transporter IRT1 by directly binding to bHLH38/101 promoter; (II) binds to the promoter regions of HIPP22/44 and activates their expression; (III) physically interacts with ABI5 and prevents its binding to the promoters of downstream genes.

**Figure 4 ijms-24-04378-f004:**
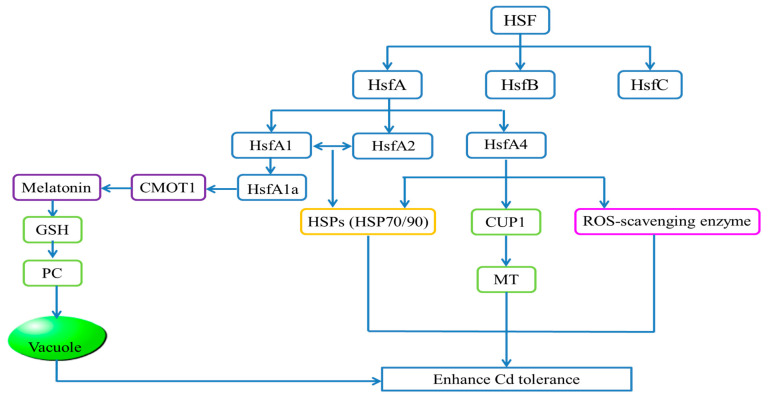
Transcriptional regulatory pathways of HSF TFs. HSFs are divided into three types: A, B, and C. HsfA plays a major role in enhancing Cd tolerance. HsfA1 interacts with HsfA2 to mediate the expression of HSPs such as HSP70 and HSP90. HsfA1a, as a kind of HsfA1, stimulates COMT1 gene expression and produces melatonin under Cd stress. Melatonin promotes the regulation of GSH and PC synthesis, causing Cd to enter the vacuole for sequestration. HsfA4 not only increases the expression of ROS-scavenging enzymes and HSPs, but also up-regulates the MT producing gene, *CUP1*, to enhance Cd tolerance.

**Table 1 ijms-24-04378-t001:** Genes involved in Cd uptake, transport, sequestration, and detoxification in plants.

Family Name	Gene Name	Main Expression Organ	Subcellular Localization	Function	Metal	Concentration	Exposure Time	Reference
ZIP family	*OsIRT1*/ *OsIRT2*	Root	Plasma membrane	Cd absorption by root	Cd/Fe	300 μM CdCl_2_/ 0.1 mM Fe-EDTA	10 d/7 d	[19,20]
	*OsZIP1*	Root	Endoplasmic reticulum, plasma membrane	Cd and Zn transport	Cd/Zn	5 μM CdCl_2_/ 12 μM ZnCl_2_	6 d/21 d	[21,22]
	*OsZIP3*	Stem	Plasma membrane	Cd accumulation	Cd/Zn	10 μM CdSO_4_/ 12 μM ZnCl_2_	7 d/21 d	[21,23]
	*OsZIP6*	Root, stem	Plasma membrane	Cd transport	Cd/Zn	0.05 μM CdCl_2_/ 1, 20 μM ZnCl_2_	21 d	[24]
	*OsZIP7*	Root, node	Plasma membrane	Cd and Zn accumulation	Cd/Zn	0.1, 0.4, 40 μM CdSO_4_/ 0.1, 0.4, 40 μM ZnSO_4_	7–28 d/ 7–28 d	[25]
	*OsZIP9*	Root	Plasma membrane	Cd and Zn uptake	Cd/Zn	5 μM CdSO_4_/ 0.04, 0.4 μM ZnCl_2_	24 h/21 d	[26,27]
NRAMP family	*AtNRAMP* *1*	Root, leaf	Plasma membrane, tonoplast	Cd uptake	Cd/Fe	2 μM CdSO_4_/0.2 mM FeCl_3_	14 d/3 d	[28,29]
	*AtNRAMP* *3*	Root, leaf	Tonoplast	Cd uptake	Cd/Fe	1, 10 μM CdCl_2_/0.2 mM FeCl_3_	3 d	[29,30]
	*AtNRAMP* *4*	Root, leaf	Tonoplast	Cd uptake	Cd/Fe	500 nM CdCl_2_/0.2 mM FeCl_3_	14 d/3 d	[29,31]
	*HvNRAMP5*	Root	Plasma membrane	Cd transport	Cd/Fe/Mn	0.1, 0.5, 1 μM CdSO_4_/0.1, 2, 10 μM FeSO_4_/0.05, 0.5, 5 μM MnCl_2_	14 d	[32]
	*OsNRAMP* *1*	Root, leaf	Plasma membrane	Cd absorption by root	Cd/ Mn	0.1, 1 μM CdCl_2_/0.5, 5, 20, 80 μM Mn	3 d/7 d	[33]
	*OsNRAMP* *2*	Shoot	Tonoplast	Cd transport and accumulation	Cd	5 μM CdCl_2_	1–5 d	[34]
	*OsNRAMP* *5*	Root	Plasma membrane	Cd transport into vascular bundles	Cd/Fe/Mn	100 nM CdSO_4_/ 5, 20 μM Fe-EDTA/ 2,4,6 μM Mn	21 d/ 14 d/ 18 d	[11,33,35]
HIR family	*OsHIR1*	—	Plasma membrane and nucleus	Cd uptake	Cd/As	50 μM CdSO_4_/ 150 μM As (V)	12 d	[36]
CaCA family	*OsCDT1*/ *OsCCX2*	Node	Plasma membrane	Cd loading in xylem	Cd/Ca	0.1, 100 μM CdCl_2_/50 mM CaCl_2_	32 h, 7 d/ 3 d	[37,38]
P-type ATPase family	*AtHMA2*	Root, Stem, leaf	Plasma membrane	Cd root-to-shoot translocation	Cd	0.06, 0.15, 0.3 Cd	14 d	[39]
	*AtHMA3*	Root, shoot	Tonoplast	Cd sequestrating in vacuoles	Cd	30 μM CdCl_2_	11 d	[40]
	*AtHMA4*	Root, Stem, leaf	Plasma membrane	Cd and Zn root-to-shoot translocation	Cd/Zn/Co	40 μM CdCl_2_/3, 200 μM ZnSO_4_/40 μM CoCl_2_	24 h	[41]
	*OsHMA2*	Root, node	Plasma membrane	Cd loading in xylem	Cd	0.2, 1 μM CdCl_2_	10 d	[42]
	*OsHMA3*	Root	Tonoplast	Transportation of Cd from cytoplasm to vacuoles	Cd	0.1, 1 μM CdSO_4_	8 d	[43,44]
	*OsHMA9*	Leaf	Plasma membrane	Cd efflux	Cd	500 μM CdCl_2_	12 d	[45]
LCT transporter	*OsLCT1*	Node	Plasma membrane	Cd transporter in phloem	Cd	0.2 μM CdCl_2_	6 h, 60 d	[46,47]
MFS superfamily	*OsCd1*	Root, grain	Plasma membrane	Cd uptake in roots and accumulation in grains	Cd	1 μM CdCl_2_	20 d	[48]
ABC transporter	*OsABCG36*	Root	Plasma membrane	Cd efflux	Cd	0.1, 1, 5 μM CdSO_4_	14 d	[49]
	*AtPDR8*	Root, leaf	Plasma membrane	Cd efflux	Cd/Pb	5, 10, 20, 30 μM CdCl_2_/0.5 mM Pb(NO_3_)_2_	14–21 d	[50]
—	*CAL1*	Root	Cell membrane	Cd accumulation in leaves	Cd	10 μM CdCl_2_	7 d	[51]
PCR family	*SaPCR2*	Root	Plasma membrane	Cd efflux	Cd	10, 15, 30 μM CdCl_2_	7 d	[52]
OPT family	*OPT3*	Root, grain	Plasma membrane	Cd transporter in phloem	Cd	50 μM CdCl_2_	14 d	[53]

## Data Availability

Data sharing is not applicable to this article as no new data were created or analyzed in this study.

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
