# Peer review of "Transcriptional Regulatory Network of Plant Cadmium Stress Response"

_ijms, 2023, doi:10.3390/ijms24054378_

Round 1

Reviewer 1 Report

Having studied this article, I want to note that it needs to be radically redone. My recommendations to the authors: the first option is to dedicate the article to the effect of cadmium specifically on rice plants and refer to other plants in the context; the second option is to significantly expand the review and describe different strategies in different plants in interaction with cadmium (in this article, the second chapter is devoted only to the example of rice plants). Some aspects that I also paid attention to:

1.       The second chapter of the article «Cd Transport and Accumulation in Plants» describes the accumulation and transport of cadmium using only rice as an example (only one sentence about Arabidopsis - lines 119-120)

2.       The introduction part could be improved by establishing narrative links between sentences and paragraphs so that the reader is able to follow your argument. In particular, to make it more logical to move from the harm of cadmium against plants to harm against humans.

3.       The title of the article does not correspond to the full content of the article and almost completely echoes the title of Chapter 3 « The Transcriptional Regulatory Network in the Response to Cd Stress».

4.       Key words need to be added and transcription factors and plants indicate in the plural

5.       In table 1, only rice genes are marked, which does not correspond to the name of the table

6.       Chapter 3 «The Transcriptional Regulatory Network in the Response to Cd Stress». It is necessary to expand and structure

7.       Genes should be marked in italics

8.       The list of references must be double-checked, for example, numbers 9, 10. Many sources are missing doi

Author Response

Response Letter

To the Editor of International Journal of Molecular Sciences

Correspondence:

Yanfei Ding: [email protected]

Dear editor,

First, we would like to thank you for your review of our manuscript. According to the reviewers’ the valuable comments and your kind suggestions, we have carefully revised our manuscript. All changes made throughout the revised manuscript are in red color. We also explain point-by-point the changes made in response to your suggestions and the reviewers’ comments.

If you have any question about the revised paper, please don’t hesitate to let me know. Once again, we acknowledge your comments and constructive suggestions very much. We hope this revision will be accepted for publication in your journal. Thank you very much.

Sincerely yours,

Yanfei Ding

Response to reviewer 1

Major comments

Having studied this article, I want to note that it needs to be radically redone. My recommendations to the authors: the first option is to dedicate the article to the effect of cadmium specifically on rice plants and refer to other plants in the context; the second option is to significantly expand the review and describe different strategies in different plants in interaction with cadmium (in this article, the second chapter is devoted only to the example of rice plants).

Reply:Thank you very much for your valuable suggestions. I have significantly expanded the review and described different strategies in different plants in interaction with Cd, particularly in the second chapter. We have added 9 genes, such as AtNRAMP1, AtHMA2, AtPDR8 from other plants. Please see the revisions in the text.

Other comments

  1. The second chapter of the article «Cd Transport and Accumulation in Plants» describes the accumulation and transport of cadmium using only rice as an example (only one sentence about Arabidopsis - lines 119-120)

Reply:Thank you very much for your suggestion. We have added 9 genes, such as AtNRAMP1, AtHMA2, AtPDR8 from other plants. Please see the revisions in the text.

  1. The introduction part could be improved by establishing narrative links between sentences and paragraphs so that the reader is able to follow your argument. In particular, to make it more logical to move from the harm of cadmium against plants to harm against humans.

Reply:Thank you very much for your suggestion. We have rewritten the first paragraph to make it more logical. Please see the revisions in the text.

  1. The title of the article does not correspond to the full content of the article and almost completely echoes the title of Chapter 3 « The Transcriptional Regulatory Network in the Response to Cd Stress».

Reply:Thank you very much for your suggestion. Recent studies have identified many transporters involved in Cd uptake, transport and detoxification. However, the complex transcriptional regulatory networks involved in Cd response remain to be elucidated. Thus, we focus on the transcriptional regulatory network in the response to Cd Stress. The title of Chapter 3 has been changed to “The Transcriptional Regulation of Cd response by TFs”. Chapter 4 and 5 focus on DNA methylation, lncRNAs, miRNAs and kinases, which are also involved in Cd-induced transcriptional responses.

  1. Key words need to be added and transcription factors and plants indicate in the plural

Reply:Thank you very much for your suggestion. We have corrected the errors

  1. In table 1, only rice genes are marked, which does not correspond to the name of the table

Reply:Thank you very much for your suggestion. We have added 9 genes, such as AtNRAMP1, AtHMA2, AtPDR8 from other plants.

  1. Chapter 3 «The Transcriptional Regulatory Network in the Response to Cd Stress». It is necessary to expand and structure

Reply:Thank you very much for your suggestion. We have rearranged the paper. Chapter 3 “The Transcriptional Regulation of Cd response by TFs”. Chapter 4 “Regulation of Cd response by DNA Methylation, Long RNAs, and Small RNAs”. Chapter 5 “How plants sense and transduce Cd signal to transcriptional regulators”.

  1. Genes should be marked in italics

Reply:Thank you very much for your suggestion. We have corrected the errors

  1. The list of references must be double-checked, for example, numbers 9, 10. Many sources are missing doi.

Reply:Thank you very much for your suggestion. We have corrected the errors.

Reviewer 2 Report

Cadmium is an important heavy metal that is toxic and harmful to human health. With the development of industrialization and the environmental pollution of human life, more and more attention has been paid to the impact of cadmium on human health. This paper summarizes the mechanism of absorption, transport and distribution of cadmium in plants, mainly introduces the transcriptional regulation network, also introduces the epigenetic mechanism such as DNA methylation, long RNA and small RNA, and explores the Cd signal, which are significant for us to reduce the accumulation of cadmium in food, or use plants to remove the cadmium pollution in soil. This article has a correct topic selection, clear description, clear views, sufficient evidence, and can be accepted for publication. However, there are a few minor questions that readers need to reconfirm.

1.     In Table 1, OsHMA3, P-Type Heavy Metal ATPase Tonoplast. However, according to my mind, P-ATPase is located in plasmalemma, not tonoplast. Only V-ATPase or PPase located in tonoplast. Please check it.

2.      L. 112, Cd contents the Cd content

3.     L. 135, below the Chinese limit. Authors should provide literatures.

4.     L. 151, (3.9 or 1.2 mg/kg Cd). It cannot be understood.

5.     L. 154, DEF8 mutant def8 mutant.

6.     L. 238, including [65]. I cannot understand it.

7.     In references, according to my mind, the first letters of the word of the title need not be capitalized except the first.

8.     The journal names should be uniform. Either the full name, or the abbreviation. Do not have the full name, some abbreviation. It looks inconsistent.

Author Response

Response Letter

To the Editor of International Journal of Molecular Sciences

Correspondence:

Yanfei Ding: [email protected]

Dear editor,

First, we would like to thank you for your review of our manuscript. According to the reviewers’ the valuable comments and your kind suggestions, we have carefully revised our manuscript. All changes made throughout the revised manuscript are in red color. We also explain point-by-point the changes made in response to your suggestions and the reviewers’ comments.

If you have any question about the revised paper, please don’t hesitate to let me know. Once again, we acknowledge your comments and constructive suggestions very much. We hope this revision will be accepted for publication in your journal. Thank you very much.

Sincerely yours,

Yanfei Ding

Response to reviewer 2

  1. In Table 1, OsHMA3, P-Type Heavy Metal ATPase Tonoplast. However, according to my mind, P-ATPase is located in plasmalemma, not tonoplast. Only V-ATPase or PPase located in tonoplast. Please check it.

Reply:Thank you very much for your review. According to the following two articles, we found that both OsHMA3 and AtHMA3 are located in tonoplast.

1). Ueno, D.; Yamaji, N.; Kono, I.; Huang, C.F.; Ando, T.; Yano, M.; Ma, J.F. Gene Limiting Cadmium Accumulation in Rice. Proc Natl Acad Sci U.S.A. 2010, 107, 16500–16505, doi:10.1073/pnas.1005396107.

2) Morel, M.; Crouzet, J.; Gravot, A.; Auroy, P.; Leonhardt, N.; Vavasseur, A.; Richaud, P. AtHMA3, a P1B-ATPase Allowing Cd/Zn/Co/Pb Vacuolar Storage in Arabidopsis. Plant Physiol. 2009, 149, 894–904, doi:10.1104/pp.108.130294.

  1. 112, Cd contents → the Cd content

Reply:Thank you very much for your suggestion. We have corrected the errors

  1. 135, below the Chinese limit. Authors should provide literatures.

Reply:Thank you very much for your suggestion. We have added the literature and marked in red color.

  1. 151, (3.9 or 1.2 mg/kg Cd). It cannot be understood.

Reply:Thank you very much for your suggestion. We have rewritten the sentence. Please see the revisions in the text (page 9, line 155).

  1. 154, DEF8 mutant → def8 mutant.

Reply:Thank you very much for your suggestion. We have corrected the errors and marked in red color.

  1. 238, including [65]. I cannot understand it.

Reply:Thank you very much for your suggestion. We have deleted the errors and marked in red color.

  1. In references, according to my mind, the first letters of the word of the title need not be capitalized except the first.

Reply:Thank you very much for your suggestion. We found that the reference require of IJMS is the word of the title need to be capitalized.

  1. The journal names should be uniform. Either the full name, or the abbreviation. Do not have the full name, some abbreviation. It looks inconsistent.

Reply: Thank you very much for your suggestions. We have already corrected these errors. Please see the revisions in the text.

Round 2

Reviewer 1 Report

The aricle «Transcriptional Regulatory Network of Plant Cadmium Stress Response» has been corrected and the authors have accepted most of my comments.

However,

11. I would like to see in the text at least a mention of the basic strategies for the relationship of plants to the accumulation of Cadmium, namely: Accumulators, Indicators, Excluders.

22. Chapter 2.3. «Cd Transport through the Phloem to Grains». This section describes the transport of Cadmium through the phloem to grains using rice as an example. Please expand and add another plant/plants (such as the barley you described earlier or another cereal plant).

Author Response

Response Letter

To the Editor of International Journal of Molecular Sciences

Correspondence:

Yanfei Ding: [email protected]

Dear editor,

First, we would like to thank you for your review of our manuscript. According to the reviewers’ the valuable comments and your kind suggestions, we have carefully revised our manuscript. All changes made throughout the revised manuscript are in red color. We also explain point-by-point the changes made in response to your suggestions and the reviewers’ comments.

If you have any question about the revised paper, please don’t hesitate to let me know. Once again, we acknowledge your comments and constructive suggestions very much. We hope this revision will be accepted for publication in your journal. Thank you very much.

Sincerely yours,

Yanfei Ding

  1. I would like to see in the text at least a mention of the basic strategies for the relationship of plants to the accumulation of Cadmium, namely: Accumulators, Indicators, Excluders.

Reply:Thank you very much for your valuable suggestions. I have added the content and divided the plants into excluder, indicator and hyperaccumulato. Please see the revisions in the text.

  1. Chapter 2.3. «Cd Transport through the Phloem to Grains». This section describes the transport of Cadmium through the phloem to grains using rice as an example. Please expand and add another plant/plants (such as the barley you described earlier or another cereal plant).

Reply:Thank you very much for your valuable suggestions. We have added the transport of Cd through the phloem to grains in other plants such as peanut, linseed, potato, and Sedum alfredii. We also have added OPT3 gene involved in phloem Cd transport in Arabidopsis. Please see the revisions in the text.